# Corporate Governance and Financial Performance: The Interplay of Board Gender Diversity and Intellectual Capital

**Zeineb Ouni** [1,*,†] , **Jamal Ben Mansour** [2,†] and **Sana Arfaoui** [3]

1   Department of Finance and Economics, School of Management, University of Quebec in Trois-Rivières, 3351 des Forges BLVD, P.O. Box 500, Trois-Rivières, QC G9A 5H7, Canada
2   Department of Human Resources Management, School of Management, University of Quebec in Trois-Rivières, 3351 des Forges BLVD, P.O. Box 500, Trois-Rivières, QC G9A 5H7, Canada
3   School of Science Administration, TÉLUQ University, 5800 Saint-Denis St, Montréal, QC H2S 3L5, Canada
*   Correspondence: zeineb.ouni@uqtr.ca; Tel.: +1-819-376-5011 (ext. 3180)
†   These authors contributed equally to this work.

**Abstract:** Prior research has found mixed evidence regarding the relationships between board gender diversity (BGD) and firm value. Moreover, there is a lack of evidence on the channels through which BGD affects firm performance; hence, this paper tackles this issue. We aim to investigate the relationship between BGD and firm performance and to explore the mediating role of intellectual capital efficiency (ICE) in this relationship. Using a multivariate regression analysis and a sample of 4008 North American firms from 2002 to 2020 (14,382 firm-year observations), we find that gender diversity is positively related to financial performance, confirming that a diversified board improves board effectiveness and brings new resources to the firm, which allows it to improve its performance. More interestingly, the results of the Structural Equation Model (SEM) indicate that the relationship between gender diversity and performance is more pronounced with the mediating role of ICE. Our results are robust, controlling for the endogeneity and heteroscedasticity issues, with several controls for firm- and country-level characteristics, using alternative sample compositions and alternative econometric techniques, and including year, industry, country and firm-fixed effects. Interestingly, this paper shows strong evidence that the effect of BGD on firm value is more effective by incorporating the role of intellectual capital efficiency.

**Keywords:** gender diversity; intellectual capital efficiency; firm performance

**JEL Classification:** G32; G15; G34

## 1. Introduction

The definitions of terms regarding the notion of 'diversity' are constantly evolving and becoming more precise over time, according to the evolution of society and emerging challenges. Thus, it is appropriate to clarify, at the outset of this paper, what we mean by 'gender diversity in the workplace', which is at the heart of our study. First, the term 'diversity at work' does not refer to practices (or the quality of the practices) that promote the integration, inclusion or involvement of people from various (social or ethnic) backgrounds. The term 'gender diversity' does not refer to differences between a person's gender identity, role or expression and the cultural norms prescribed for people of a particular gender. In fact, we use the term 'diversity at work' to qualify a work environment according to the presence (or not) of groups of people who differ with regard to social/ethnic origin or considering other attributes such as gender, age or sexual orientation. In the same vein, the term 'gender diversity in the workplace' is limited to groups of people in a given work context (employees, managers, or board members) who self-declare as 'male' or 'female' based on either 'sex' or 'gender' attributes.

Research on 'gender diversity in the workplace' today presents a rich and varied body of knowledge. Many perspectives have been studied to examine this topic, but the most dominant is the one who is interested in the impact of gender diversity on corporate governance and firm value [1–10], etc. Indeed, in last several years, researchers, practitioners, public administrations and private companies have shown an increasing interest in corporate governance, mostly due to repeated scandals that the financial markets have experienced in Europe and North America (e.g., Enron, Vivendi, etc.). According to Shleifer and Vishny [11], "*Corporate governance deals with the ways which suppliers of finance to corporations assure themselves of getting a return on their investment*", In other words, corporate governance refers to a set of mechanisms and tools that allow better control by reducing the sources of conflict within the corporation. However, better governance is primarily based on a qualified board of directors [12,13], etc. Indeed, the role of BOD is to reduce agency costs [14], by monitoring and controlling managers to ensure that their decisions converge with the firm's objective of creating value and strengthening its social legitimacy. Several previous studies have examined the quality and effectiveness of the board of directors by studying its characteristics such as independence, size, number of meetings, networking, experience, as well as its diversity in terms of age, nationality, experience gender, etc.

Moreover, we know that the evolution of society, the intense presence of technology, the rapidity of changes in the organizational environment, the upgrading of the skills of qualified workers (knowledge workers), the importance of innovation replaces intellectual capital as a lever for both the profitability and the sustainability of the organization. Indeed, for several authors, the role of intangible capital on value creation is even more important than tangible capital. In the same vein, the resource-based theory perspective considers intangible resources the main determinant of sustainable competitive advantage. However, in the analysis of the existing research, we can notice that studies focus more on the effect of gender diversity on certain tangible results as firm value, with little interest in the role of intangible capital such as intellectual capital. (e.g., human capital, social capital, structural capital).

This study contributes to our knowledge regarding the relationship between gender diversity on boards and corporate financial performance. Specifically, our goal is to provide a better understanding and an explanation for the disparity in the results of empirical studies. First, the existing literature uses an international analytical framework, which could explain the divergent results due to the diversity in regulatory conditions and the socio-cultural context. Unlike these studies, we examine only North American firms. On the one hand, North American countries (USA and Canada) have similarities in economic and institutional conditions. Therefore, this analysis framework allows us to isolate the confounding effects of factors related to the national context that could affect the nature of the relationship between gender diversity on the board and firm performance. Nevertheless, on the other hand, Canadian and American corporate governance systems seem to be different (principle- versus rule-based approaches). Therefore, this study allows us to investigate if there are any differences into the BGD–performance relationships between Canadian and American firms. Second, our study extends the period over 18 years (2002–2020), which allows us to avoid the possibility that our results are conditional to a particular period of the company's performance. Third, this study provides an empirical contribution to the existing debate about the added value of gender diversity in corporate decision-making. Fourth, in addition to considering the well-documented effect of gender diversity on financial performance [15–18], our study examines channels through which gender diversity could affect firms' financial performance. More specifically, we aim to investigate the mediating role of "Intellectual Capital Efficiency" (henceforth ICE), which is commonly known as a source of competitive advantage in the era of rapid change, innovation and a knowledge-based economy, in the relationship between gender diversity on the BOD and financial performance.

## 2. Literature Review and Hypotheses Development

Diversity in a Board of Directors is a controversial topic. For some, diversity is above all an ethical issue. For others, diversity is important because it allows more creation of economic wealth for the company. It is likely that the introduction of a legal requirement to engage women to the board and the absence of confirmation on the effectiveness of this approach has led to the research enthusiasm for the impact of gender diversity.

In principle, diversity contributes to the understanding of the board regarding the business environment of a company by enriching and adding to their perspectives and cognitive frameworks. Indeed, gender diversity on a board would increase capability and creativity [19] and enhance the quality of information circulating within the company [20]. Several empirical studies have studied the relationship between diversity and performance. However, the results of these studies are divergent and inconclusive. Indeed, some studies show that BODs with more women positively affects firm returns on assets [21,22] and induce a positive stock market reaction [23]. In contrast, in other studies, gender diversity on the board negatively affects firm accounting performance [24,25] and reduce share value [26]. Several other studies have found no relationship between gender diversity and company performance [9,27,28]. Furthermore, Hermalin and Weisbach's [14] have tried to explore the mechanisms underlying the relationship between diversity and performance.

### 2.1. BGD and Financial Firm Performance Hypothesis

The impact of board gender diversity on firm performance remains an empirical issue [29,30]. However, for the purpose of this paper, we follow previous studies [9,31], etc., and we apply the agency, resource-based and cognitive theories to develop the conceptual framework predicting the nature of the relationship between board gender diversity and the firm performance. According to agency theory, a competent and independent board of directors would be able to fulfill its control and monitoring function and thus reduce agency conflicts between managers and shareholders, which eventually increases the firm's value [32]. The literature on corporate governance has focused on the characteristics of the board of directors to assess its quality, namely board size, board independence, board meeting attendance, board expertise, board composition, etc. For the purposes of this study, we refer to board composition, specifically board gender diversity, as a governance mechanism. Recent empirical research suggests that having female board members improves board monitoring and brings new resources to decision-making [7,33], etc.

As we have already explained in our previous paper [31], page 3 of 17, "resource-dependency theory and cognitive theories consider board members as active actors in the creation of value in several ways. For the resource-dependency theory, firms operate in an open system and are forced to exchange with their environment for the acquisition of certain resources. Thus, board members contribute in terms of reducing the risks of access to resources (e.g., skills, relationships) that are critical and indispensable for the survival of the firm. According to this theory, the effectiveness of a BD lies in its ability to monitor but also to facilitate access to resources. For cognitive theories, board members make a strategic contribution through the variety in their knowledge, cognitive cues, and perspectives. Thus, the diversity of the board members guarantees access to a diversity of cognitive frameworks, and thus to multiple readings of reality and a wealth of strategic orientations. It is from this perspective that studies on the effectiveness of a board of directors find their theoretical basis, justifying the importance of diversity, including gender diversity, on the board of directors. Indeed, several studies support the idea that heterogeneous groups are superior to homogeneous groups [31], (pp. 7–17) because of the tendency of members to engage in substantive discussions and to integrate the variety of knowledge and information available [34,35]. Adams and Funk's study of 502 board members and 126 CEOs regarding their value systems according to the Schwartz model [36–38], revealed significant male–female differences that were similar to the general population for certain aspects and different from the population for others. More specifically, female board members were more involved in volunteer work (activities that preserve and improve the

well-being of those with whom they are in frequent personal "group" contact), and were more universalist (understanding, appreciative, tolerant, and protective of the well-being of all and of nature) but less power-oriented (social status and prestige, control or domination over people and resources) than their male counterparts. However, in contrast to the general population, women on BDs had fewer traditional values (from culture and religion), less security orientation (preserving existing social arrangements), and comparable risk aversion to men". Based on the above arguments, the first hypothesis suggests that board gender diversity could represent a good governance mechanism resulting in a higher financial performance. We state our first hypothesis as follows:

**Hypothesis 1 (H1):** *Board gender diversity positively affects the firm's financial performance.*

*2.2. BGD and Intellectual Capital Efficiency Hypothesis*

The term '*human capital*' (HC) is used to refer to the totality of an organization's human capital (e.g., training, expertise), structural capital (e.g., processes, culture) and social capital (e.g., legitimacy, relationship with the community). Today, knowledge-based economy and the transformation of work require more skills from employees (knowledge worker) and replace human capital back at the center of organizations' concerns. The importance of innovation for organizations' sustainability and competitiveness invites them to reconsider their internal processes, their culture, the organization of work, and, therefore, their structural capital. The organization's status, its relationship with the community, social commitment, legitimacy, etc., are strategic resources that are part of social capital. In other words, intellectual capital is a stock of resources necessary for the development of an organization's distributive (e.g., brand, process) and reproductive (e.g., employee skills) capabilities, competitiveness, value creation and profitability.

Based on Hambrick and Mason's [39] '*upper-echelon theory*', it is accepted that it is the internal characteristics of the highest hierarchical level of an organization that determines, in part, its results. Applied to a BOD, several studies reveal that diversity enhances innovation through strategic orientations and innovative decisions [40], which improves structural capital. Talke et al. [41] recognize the importance of diversity for the flow of quality information, flexibility and predisposition to adopt innovations, which helps organizations choose directions that facilitate the introduction of innovative ideas into their services and products.

Empirical studies document at least four arguments for the importance of gender diversity, namely performance, governance, market legitimacy and access to talent [42]. Low et al. [43] document that gender diversity plays a very important role in corporate governance. In addition, it makes human and structural resources more efficient and improves business performance and stakeholder management. The efficiency of instinctual capital is also a question of managing the whole communication aspect with the organizations' external environment, and gender diversity contributes to this according to other studies [44,45]. The role of gender diversity is also supported with regard to the relational aspect between people who constitute the human capital of any organization [46], the performance aspect of employees [47], and the informational aspect in terms of market sharing, customers and strategic challenges [48] in order to support creativity and innovation [44,49]. With reference to these arguments, the second hypothesis suggests that board gender diversity as a governance mechanism improves intellectual capital efficiency. We state our second hypothesis as follows:

**Hypothesis 2 (H2):** *Board gender diversity positively affects the intellectual capital efficiency.*

*2.3. Intellectual Capital Efficiency and Firm Performance Hypothesis*

At this phase of the manuscript, we have advanced two hypotheses, one linking gender diversity in BODs with performance (H1) and the other linking gender diversity in BODs with the efficiency of intellectual capital (H2). In order to complete our analysis, it is rea-

sonable to adopt the idea of mediation of efficiency of intellectual capital to explain the first relationship. Before constructing this hypothesis, let us first recall that through its content in terms of employees' knowledge and skills, applied experience, processes, policies, presence of technology, work environment, relationship with stakeholders including customers and suppliers, innovation produced, and brands, intellectual capital (in its human, structural or relational dimension) is an important factor in a service-oriented knowledge economy. The relationship between intellectual capital and organizational performance could be explained by several theories. For example, the "Resource-Based Theory" from the field of strategy, stipulates that an organization's resources, particularly intangible resources, provide a sustainable competitive advantage. Boudreau and Ramstard's 'HR Bridge Framework' [50] from the field of human resources management, or Walters's model [51] from the field of finance have also discussed this relationship. However, despite the obvious role of intellectual capital (IC), its impact on the value of the company is not widely understood among practitioners for at least three reasons. First, conceptually, it is difficult to measure, let alone isolate, the effect of IC on financial performance. Economically, it takes time and money to find any information that is intellectual capital. Strategically, the content and value of its IC could be used by the competition [52]. Previous research has successfully developed the VAIC index, which is an important measure of IC combining human capital efficiency, structural capital efficiency, and physical and financial capital (Capital Employed Efficiency). Thus, it becomes simpler to measure and isolate the effect of each component of intellectual capital on financial performance. Thus, several studies have shown a positive and significant relationship between intellectual capital (or one of its components) and performance, but this relationship takes on many patterns. For example, Mavridis's study [34] shows a relationship only between human capital and performance among Japanese banks. In another study of Malaysian firms, Bontis et al. [35] show a direct relationship only between structural capital and performance. However, this same study reveals an indirect effect of human capital on structural capital, which leads to the conclusion that, overall, intellectual capital has an effect on performance. Another German study [53] concludes that all components of intellectual capital indirectly influence performance through intellectual property (intellectual property, IP). The inconsistency of the results does not allow us to draw a conclusion about the relationship between intellectual capital and firm performance. The first explanation lies in the variety of scales measuring intellectual capital, while the second explanation refers to a variety of contexts of the companies studied. Now, limiting the literature review to only those studies that chose the VAIC index eliminates difficulties arising from the variety of scales measuring intellectual capital. For example, studies such as Chen et al., Shiu, and Wei et al. [16,36,37] found a significant relationship between VAIC, HCE, and CEE and performance as measured by the ROA. Chen et al. [16] study adds three other indicators of performance, Return on Equity, Revenue Growth, and Employee Productivity. Although several studies support a positive relationship between VAIC and performance, the way in which performance is measured tips the relationship in one direction or the other. Indeed, studies such as Firer and Williams and Shiu [36,38] reveal a negative relationship between human capital (HCE) and firm performance as measured by indicators such as Asset turnover and Market to Book ratio. In other studies, such as Appuhami [54], the relationship with performance measured with the capital gains made by the investor's index is positive but not significant. As for structural capital (SCE), the relationship with performance (ROA) is very weakly supported with a few exceptions [16,38,55], and in some cases, this relationship is even negative [37]. For CEE, several studies support a positive relationship with performance, but there are some that show a negative relationship [54]. Based on the above arguments, the third hypothesis suggests that intellectual capital efficiency improves firm performance. We state our third and fourth hypotheses as follows:

**Hypothesis 3 (H3):** *The intellectual capital efficiency positively affects the firm's financial performance.*

**Hypothesis 4 (H4):** *The intellectual capital efficiency mediates the relationship between board gender diversity and firm's financial performance.*

## 3. Research Methods

In this section, we describe the procedure of sample selection, present the variables and empirical specification used in the analysis and report descriptive statistics.

### 3.1. Sample Selection and Data Sources

The sample consists of North American publicly traded firms (USA and Canada), and we collected our data from several sources. In fact, most of the research on the effect of corporate governance on corporate performance have taken as a framework of analysis the USA firms, considering that the results could be generalized for Canadian firms. This is justified by the fact that the two neighboring countries have certain resemblances in terms of the legal and economic system, financial regulations, etc. However, despite these similarities, corporate governance regulation seems to be different between the two countries. Indeed, the corporate governance system in Canada can be defined as a "principles-based" approach, while the U.S. governance system can be defined as a "rules-based" approach [56–58]. The differences in governance systems could affect Canadian and American firms differently. Thus, this study will examine whether there are any differences in the relationships of gender diversity on performance, specifically between Canadian and American firms. To construct our sample, our starting point is the Asset4 ESG database, which has provided data on gender diversity and corporate governance since 2002 (it is worth mentioning that the sample period starts in 2002, which coincides with the passing of the Sarbanes–Oxley Act). We retrieved diversity data for all U.S. and Canadian publicly traded firms and for the entire period covered by the database. After, we collected financials and the firm's intellectual efficiency data (ROA, size, leverage, risk, etc.) from Worldscope and finally economic variables from World Competitiveness Center Database. We then merged these different data sources and eliminated firms with missing financial, governance or economic data. Our final sample consists of 14,382 firm-year observations covering 4008 North American firms (3556 U.S. firms and 452 Canadians firms) for the period 2002–2020. The sample period gives us the opportunity to examine the Diversity–Performance–ICE relationship over a long period and to take into account the evolution of the integration of laws and regulations concerning diversity within firms. Appendix B reports our sampling process.

### 3.2. Regression Variables

As existing literature [7,59], etc. we use return on assets (ROA) as our principal proxy for firm operating profitability, which is calculated as "the ratio of earnings before interest, taxes, depreciation and amortization to total assets". For robustness, we used alternative proxies for performance, such as return on equity (ROE), return on sales (ROS) and Tobin's Q.

As a measure for board gender diversity and following previous research, e.g., [27,54–58], *BGD* is defined as the percentage of female representatives on the board. For robustness, we used the percentage of female executive members (ExMemGD (%)) and the percentage of female managers compared to the total number of managers (WomMangScore) as other measures for board gender diversity.

Consistent with the literature on intellectual capital efficiency, e.g., [60–62], we use the adjusted-value-added intellectual capital (IC) coefficient model (*A-VAIC*) as a composite measure of human, innovation and financial or physical capital as a proxy for IC efficiency. Appendix A provides the method of calculating these coefficients based on data from Datastream database.

To avoid omitted variable bias, we control for: (i) firm size (Size) measured by the natural logarithm of total assets. Larger firms are more visible, have a better reputation and have higher credit quality; (ii) Leverage as a proxy for the firm's financial vulnerability

and measured by the total debts to total assets. Firms with higher degrees of Leverage would face a higher risk of liquidity; (iii) Firm age (Age) as proxy for the experience and expertise of the firm. We calculated Age as the difference between the specific year of performance data and the firm year incorporation; (iv) Altman's Z-score (Z-Score) as a proxy for the firm's likelihood of bankruptcy. We also include CAPEX as a proxy for firm growth opportunities (measured as the ratio of capital expenditures to total assets). Firm ownership (Held Shares) represents shares held by insiders. We also added proxies for firm governance quality (Gov and CEO duality). Finally, we use the GDP growth as a measure variable for macroeconomic conditions. However, and to assess the sensitivity of our results, we try to collect additional data such as BoardSize, Loss, Beta, Inflation, firm policy on board diversity, etc. Appendix A provides descriptions and data sources for all the variables. To alleviate the effect of outlier observations, we note that all continuous variables are winsorized at the 1 percent and 99 percent levels. In addition, we lagged by one year all controls variables to mitigate simultaneity concerns.

### 3.3. Empirical Specifications

To study the relationship between the board gender diversity and firm performance, we use the following general specification:

$$
\begin{aligned}
FIRM\ PERFORMANCE_{i,t} = {} & \beta_0 + \beta_1\ GBD_{i,t-1} + \beta_2\ Gov_{i,t-1} + \beta_3\ CEO\ duality_{i,t-1} + \beta_4\ Held\ Shares_{i,t-1} + \\
& \beta_5\ CAPEX_{i,t-1} + \beta_6\ Leverage_{i,t-1} + \beta_7\ Age_{i,t-1} + \beta_8\ Z\text{-}Score_{i,t-1} + \beta_9\ Size_{i,t-1} + \beta_{10}\ GDP_{i,t-1} + \\
& Country\ Fixed\ Effects + Industry\ Fixed\ Effects + Year\ Fixed\ Effects + \varepsilon
\end{aligned}
\tag{1}
$$

where the dependent variable, FIRM PERFORMANCE, is the return on assets (ROA). The percentage of females on the board (BGD) is our main explanatory variable. The control variables are Gov, CEO duality, Held Shares, CAPEX, Leverage, Age, Z-Score, Size and GDP. $\varepsilon$ is an error term. Our main interest in the analysis is the BGD coefficient estimate, $\beta 1$. Positive coefficients, for instance, mean that firm performance increases with board gender diversity.

Following prior research, see, for instance, [31], we use the ordinary least square (OLS) method to estimate our model. We use a standard error estimation procedure to control for heteroscedasticity and account for time-series dependence. Additionally, and as suggested by Petersen and Thompson [63,64], we clustered standard errors at the firm, country, year, and industry levels. In addition, we added dummy variables to account for the country, time, and industry fixed effects and to control for unobserved factors that may influence firm performance.

Otherwise, we use a structural equation model (SEM) to investigate if intellectual capital efficiency acts as a mediating variable in the relationship between BGD and firm performance. Traditionally, the mediation effect is measured by a series of linear regression models as defined by Baron and Kenny [65]. However, the use of SEM allows us to measure one model that estimates the direct, indirect and total effects.

## 4. Analysis and Findings

### 4.1. Descriptive Analysis and Findings

Table 1 reports sample distribution. Panel A show that US-based firms dominate the sample (91.3%). These results are expected, since the American financial market is the most developed around the world, in addition to the US being a pioneering country in the implementation of regulations regarding good governance. Panel B of Table 1 reports the sample distribution by sector and according to Panel C, 54 percent of observations cover the period between 2016 and 2019. We can also notice that the period of financial crisis (2007–2008) represents only 6.5 percent of the sample, which reduces any bias due to that distress period.

**Table 1.** Sample Description (Main evidence: Basic Model). This table provides sample distributions by country, years, and industry. The sample includes 14,382 firm-year observations representing 4008 North American listed firms over the period 2002–2020. Appendix A reports all variables description.

| Panel A: Sample Distribution by Target Country | | | |
|---|---|---|---|
| **Nation** | **Freq.** | **Percent** | **Cum.** |
| Canada | 1276 | 8.87 | 8.87 |
| United States of America | 13,106 | 91.13 | 100.00 |
| Total | 14,382 | 100.00 | |

| Panel B: Sample Distribution by Target Industry | | | |
|---|---|---|---|
| **Industry Group Name** | **Freq.** | **Percent** | **Cum.** |
| Aerospace | 72 | 0.50 | 0.50 |
| Apparel | 124 | 0.86 | 1.36 |
| Automotive | 214 | 1.49 | 2.85 |
| Beverages | 108 | 0.75 | 3.60 |
| Chemicals | 458 | 3.18 | 6.79 |
| Construction | 459 | 3.19 | 9.98 |
| Diversified | 243 | 1.69 | 11.67 |
| Drugs, cosmetics and health care | 1308 | 9.09 | 20.76 |
| Electrical | 230 | 1.60 | 22.36 |
| Electronics | 1852 | 12.88 | 35.24 |
| Financial | 962 | 6.69 | 41.93 |
| Food | 357 | 2.48 | 44.41 |
| Machinery and equipment | 537 | 3.73 | 48.14 |
| Metal producers | 482 | 3.35 | 51.49 |
| Metal product manufacturers | 243 | 1.69 | 53.18 |
| Miscellaneous | 2824 | 19.64 | 72.82 |
| Oil, gas, coal and related services | 849 | 5.90 | 78.72 |
| Paper | 160 | 1.11 | 79.84 |
| Printing and publishing | 149 | 1.04 | 80.87 |
| Recreation | 507 | 3.53 | 84.40 |
| Retailers | 800 | 5.56 | 89.96 |
| Textiles | 65 | 0.45 | 90.41 |
| Tobacco | 19 | 0.13 | 90.54 |
| Transportation | 435 | 3.02 | 93.57 |
| Utilities | 925 | 6.43 | 100.00 |
| Total | 14,382 | 100.00 | |

| Panel C: Sample Distribution by Year | | | |
|---|---|---|---|
| **Year** | **Freq.** | **Percent** | **Cum.** |
| 2002 | 184 | 1.28 | 1.28 |
| 2003 | 189 | 1.31 | 2.59 |
| 2004 | 288 | 2.00 | 4.60 |
| 2005 | 354 | 2.46 | 7.06 |
| 2006 | 344 | 2.39 | 9.45 |
| 2007 | 378 | 2.63 | 12.08 |
| 2008 | 504 | 3.50 | 15.58 |
| 2011 | 693 | 4.82 | 20.40 |
| 2012 | 702 | 4.88 | 25.28 |
| 2013 | 704 | 4.90 | 30.18 |
| 2014 | 748 | 5.20 | 35.38 |
| 2015 | 1137 | 7.91 | 43.28 |
| 2016 | 1456 | 10.12 | 53.41 |
| 2017 | 1831 | 12.73 | 66.14 |
| 2018 | 2135 | 14.84 | 80.98 |
| 2019 | 2351 | 16.35 | 97.33 |
| 2020 | 384 | 2.67 | 100.00 |
| Total | 14,382 | 100.00 | |

Table 2 shows summary statistics for our key selected variables during the sample period. In terms of profitability, as mainly measured by the ratio of return on assets (ROA), the average is about 2.87 percent, which is in agreement with previous research [7]. Moreover, the average of BGD is 16.37 percent, which is relatively low, but consistent with prior studies [33] and the Catalyst global census of women on boards, which indicates that in 2014, women held approximately 19.2 percent (20.8 percent) of board director seats of S&P 500 (Canadian) companies. This percentage improved in the USA to reach 24 percent in 2019 and 29 percent in 2022 according to Moody's Investors Service (https://www.catalyst.org/research/2014-catalyst-census-women-board-directors/ (accessed on 15 June 2022)) (Catalyst is a global non-profit organization, which provides information about the number of women board members in public firms covering 20 countries). The average percentage of governance quality is about 48.86 percent, with a maximum of 91.38 percent, which is expected considering that the USA and Canada are considered among the countries that have well-developed corporate governance mechanisms. The mean of shares held by insiders is 11.3 percent. This proportion is about 0.052 percent, 2.21 percent, 28.08 percent, 4.29 percent, 21.89 percent and 0.15 percent for growth opportunity (CAPEX), Leverage, firm age (Age), firm Z-Score, firm size and GDP, respectively. The A-VAIC score mean is about 4.37, and the average of SEC, CEE, and HCE represents 4.370, 0.348 and 3.435, respectively. Table 3 presents Pearson correlations between our key variable (ROA) and all the other explanatory variables. In general (except for Leverage), ROA is significantly correlated with BGD and all other independent variables with the predicted direction. Moreover, the correlation analysis indicates a weak association between the variables, which proves that there are no severe issues of multicollinearity. In addition, VIF (variance inflation factor) analysis was used and for the variables used in our main model, the highest VIF detected is about 1.30, which allows us to confirm that our variables are independent of each other.

**Table 2.** Summary statistics. This table provides summary statistics. The sample includes 14,382 firm-year observations representing 4008 North American listed firms over the period of 2002–2020. Appendix A reports all variables description.

|  | **Mean** | **Median** | **SD** | **Min** | **Max** |
|---|---|---|---|---|---|
| ROA | 2.872 | 5.270 | 14.673 | −75.820 | 31.840 |
| BGD | 16.372 | 15.385 | 10.947 | 0.000 | 50.000 |
| Gov | 48.867 | 49.589 | 22.052 | 3.654 | 91.386 |
| CEO duality | 0.976 | 1.000 | 0.153 | 0.000 | 1.000 |
| Held Shares | 11.306 | 3.100 | 16.826 | 0.000 | 99.640 |
| CAPEX | 0.052 | 0.036 | 0.053 | 0.000 | 0.299 |
| Leverage | 2.214 | 1.300 | 3.618 | 0.001 | 49.787 |
| Age | 28.084 | 20.000 | 26.737 | 0.000 | 119.000 |
| Z-Score | 4.298 | 3.002 | 5.516 | −7.495 | 38.182 |
| Size | 21.892 | 21.938 | 1.720 | 17.250 | 26.906 |
| GDP | 0.153 | 0.231 | 0.349 | −2.438 | 0.766 |
| SEC | 4.370 | 3.022 | 4.974 | −3.359 | 18.776 |
| CEE | 0.348 | 0.249 | 0.414 | −0.653 | 1.955 |
| HCE | 3.435 | 1.926 | 4.112 | −2.855 | 18.064 |
| VAIC | 4.541 | 2.644 | 5.542 | −4.011 | 22.634 |

**Table 3.** Pearson Correlations. This table reports Pearson's correlation coefficients. The sample includes 14,382 firm-year observations representing 4008 North American listed firms over the period 2002–2020. The superscript ***, **, and * denote statistical significance at the 1%, 5%, and 10% levels, respectively. Appendix A reports all variables' descriptions.

| Variables | (1) | (2) | (3) | (4) | (5) | (6) | (7) | (8) | (9) | (10) | (11) |
|---|---|---|---|---|---|---|---|---|---|---|---|
| (1) ROA | 1.000 | | | | | | | | | | |
| (2) BGD | 0.076 *** | 1.000 | | | | | | | | | |
| (3) Gov | 0.186 *** | 0.309 *** | 1.000 | | | | | | | | |
| (4) CEO duality | −0.022 *** | 0.027 *** | 0.031 *** | 1.000 | | | | | | | |
| (5) Held Shares | −0.086 *** | −0.147 *** | −0.298 *** | −0.087 *** | 1.000 | | | | | | |
| (6) CAPEX | 0.029 *** | −0.096 *** | 0.028 *** | −0.014 * | 0.034 *** | 1.000 | | | | | |
| (7) Leverage | 0.000 | 0.046 *** | 0.013 * | −0.018 ** | 0.011 | −0.016 * | 1.000 | | | | |
| (8) Age | 0.158 *** | 0.144 *** | 0.156 *** | 0.032 *** | −0.084 *** | −0.071 *** | 0.011 | 1.000 | | | |
| (9) Z-Score | 0.125 *** | −0.044 *** | −0.085 *** | 0.005 | 0.065 *** | −0.050 *** | −0.194 *** | −0.022 *** | 1.000 | | |
| (10) Size | 0.344 *** | 0.205 *** | 0.351 *** | 0.019 ** | −0.226 *** | 0.043 *** | 0.148 *** | 0.195 *** | −0.235 *** | 1.000 | |
| (11) GDP | −0.022 *** | 0.056 *** | −0.004 | −0.007 | 0.016 * | −0.014 * | −0.007 | −0.026 *** | 0.012 | −0.056 *** | 1.000 |

### 4.2. Inferential Analysis and Findings

This section presents our main results. We begin by documenting the empirical association between our main measure of performance (ROA) and the percentage of females on the board. We then present our robustness checks for addressing endogeneity concerns and report results of some alternative specifications and empirical approaches.

#### 4.2.1. The Impact of Board Gender Diversity on Firm Performance: Main Evidence

Panel A of Table 4 presents the results of differences in ROA means. The average ROA for firms with a policy of board gender diversity is higher than the average ROA of firms without a policy of board gender diversity. The difference between the ROA of the two groups (about 15.18%) is very significant, meaning that having a policy of board gender diversity increases firm value. To go deeper in our analysis, we now perform our multivariate regression model.

**Table 4.** ROA mean difference tests. This table reports various comparison tests. Panel A report *t*-tests results. Panel B reports comparison tests using Propensity Score Matching. Appendix A reports all variables description.

| Panel A: ROA Mean—Comparisons Test | | | | | Panel B: ROA Mean—Comparisons Test Using Propensity Score Matching | | | | |
|---|---|---|---|---|---|---|---|---|---|
| Group | N | Mean | t-Statistic | Sig. [Pr(T < t)] | Group | N | Difference: (1)–(2) | t-Statistic | Sig. [Pr(T < t)] |
| (1) Policy Board Diversity = 0 | 2377 | −12.32 | −22.69 | 0.000 | (1) Treatment group | 7204 | 13.39 | 11.59 | 0.000 |
| (2) Policy Board Diversity = 1 | 7563 | 2.857 | | | (2) Control group | 2313 | | | |
| Combined | 9940 | −0.774 | | | | | | | |
| Difference: (1)–(2) | | −15.18 | | | | | | | |

Table 5 presents the results of our multivariate regressions. We consider several specifications, all of which are ordinary least squares (OLS) regressions. We use standard errors corrected for heteroscedasticity, clustered at the firm, industry, country and year level and introduce dummies to control for the year, country and industry fixed effects.

**Table 5.** The impact of Gender Diversity on Firm Performance: Main Evidence and Robustness Checks. This table presents our main evidence on the impact of gender diversity on firm performance. The main dependent variable is the firm value measured by ROA. In Model (4), we control for the firm's fixed effects. For robustness, other measures were used, such as ROE in Model (5), TobinQ in Model (6), and ROS in Model (7). Gender diversity is measured by the percentage of women on the board. For robustness, gender diversity is measured by percentage of female executive members (ExMemGD (%)) and the percentage of female managers to the total number of managers (WomMangScore). The main sample includes 14,382 firm-year observations representing 4008 listed North American firms over the period of 2002–2020. Appendix A reports all variables description. The superscript ***, **, and * denote statistical significance at the 1%, 5%, and 10% levels, respectively.

| | (1) Dep Var: ROA | (2) Dep Var: ROA | (3) Dep Var: ROA | (4) Dep Var: ROA | (5) Dep Var: ROE | (6)DepVar: Tobin-Q | (7) Dep Var: ROS | (8) DepVar: ROA | (9) Dep Var: ROA |
|---|---|---|---|---|---|---|---|---|---|
| BGD | 0.176 *** | 0.025 *** | 0.024 ** | 0.013 *** | 0.149 *** | 0.032 *** | 0.006 *** | | |
| | (6.023) | (2.63) | (2.134) | (13.157) | (6.331) | (17.602) | (13.133) | | |
| ExMemGD (%) | | | | | | | | 0.089 *** | |
| | | | | | | | | (8.583) | |
| WomMangScore | | | | | | | | | 0.019 ** |
| | | | | | | | | | (20.334) |
| Gov | | 0.051 *** | 0.053 *** | −0.008 *** | 0.121 *** | −0.002 | 0.001 *** | 0.069 *** | 0.004 |
| | | (10.056) | (7.994) | (−6.121) | (9.6) | (−1.124) | (6.785) | (7.249) | (0.484) |
| CEO duality | | −3.013 *** | −3.219 *** | −0.248 | −7.483 *** | −0.135 ** | −0.46 *** | −4.342 *** | −2.91 *** |
| | | (−7.106) | (−9.319) | (−1.067) | (−10.607) | (−2.239) | (−3.409) | (−12.229) | (−3.235) |
| Held Shares | | 0.006 | 0.009 | −0.047 ** | −0.041 * | 0 | 0 | 0.001 | −0.014 |
| | | (0.875) | (1.168) | (−3.352) | (−1.72) | (0.266) | (0.599) | (0.171) | (−0.569) |
| CAPEX | | 4.997 | 4.734 | 11.577 ** | −9.801 | −0.026 | 0.351 *** | 8.041 ** | −25.845 *** |
| | | (1.587) | (1.463) | (3.552) | (−1.325) | (−0.204) | (6.334) | (2.574) | (−4.544) |
| Leverage | | −0.013 | −0.012 | 0.14 *** | 0.895 ** | 0.024 *** | −0.005 *** | −0.107 *** | 0.206 *** |
| | | (−0.604) | (−0.415) | (15.43) | (2.211) | (5.85) | (−4.29) | (−4.8) | (10.145) |
| Age | | 0.043 *** | 0.046 *** | 0.299 *** | 0.122 *** | −0.002 | 0.001 *** | 0.047 *** | 0.016 *** |
| | | (11.281) | (12.215) | (44.28) | (11.95) | (−1.007) | (6.925) | (14.602) | (3.799) |
| Z-Score | | 0.559 *** | 0.556 *** | 0.236 *** | 1.463 *** | 0.156 *** | 0.009 *** | 0.553 *** | 1.121 *** |
| | | (13.48) | (19.52) | (34.016) | (18.976) | (13.549) | (4.243) | (19.44) | (14.749) |
| Size | | 2.605 *** | 2.754 *** | −1.567 *** | 6.084 *** | −0.149 *** | 0.078 *** | 2.816 *** | 0.691 *** |
| | | (29.057) | (35.257) | (−26.499) | (26.172) | (−8.707) | (30.679) | (27.651) | (6.613) |
| GDP | | | −0.728 | 1.206*** | −1.468 | 0.557 *** | −0.005 | −0.109 | −1.019 |
| | | | (−1.155) | (6.455) | (−1.361) | (11.158) | (−0.302) | (−0.325) | (−1.049) |
| _cons | 2.601 *** | −55.768 *** | −58.89 *** | −12.212 *** | −131.631 *** | 4.849 *** | −1.934 *** | −60.262 *** | −7.52 *** |
| | (8.752) | (−22.312) | (−30.753) | (−7.306) | (−17.971) | (12.512) | (−36.243) | (−15.557) | (−9.061) |
| Observations | 23,200 | 15,986 | 14,382 | 14,382 | 14,300 | 4759 | 12,118 | 13,190 | 1963 |
| Adj R2 | 0.079 | 0.198 | 0.201 | 0.726 | 0.168 | 0.307 | 0.155 | 0.224 | 0.278 |
| Sig. | (0.000) | (0.000) | (0.000) | (0.000) | (0.000) | (0.000) | (0.000) | (0.000) | (0.000) |
| Year Fixed Effect | YES | YES | YES | YES | YES | YES | YES | YES | YES |
| Industry Fixed Effects | YES | YES | YES | YES | YES | YES | YES | YES | YES |
| Country Fixed Effects | YES | YES | YES | YES | YES | YES | YES | YES | YES |
| Firm Fixed Effects | NO | NO | NO | YES | NO | NO | NO | NO | NO |

Columns (1)–(3) of Table 5 report our main evidence on the impact of *BGD* on firm performance. In column (1), we present the basic model in which we do not integrate any control variables. Confirming our first hypothesis (H1), and consistently with Mohsni et al., Ouni et al., Triana et al. [31,66,67], etc., our regression analysis shows that gender diversity positively and significantly impacts firm value. Specifically, an increase of 1% in the level of board gender diversity leads to an increase in the firm's value by approximately 0.176%. This suggests that gender-diverse boards invest more in monitoring and control activities, which reduces conflicts and agency costs and then increases the firm's performance. This relationship remains significant even after we added firm controls (column (2)) and country controls (Colum (3)). Furthermore, all control variables have their expected sign except for the GDP growth rate variable. In column (3), and as expected, the profitability ratio (ROA) is significantly and positively affected by the firm: *(i)* quality governance (Gov); *(ii)* experience (Age); *(iii)* size (Size) and financial stability (Z-Score). Moreover, CEO duality negatively and significantly affects the firm's performance. However, the coefficients of Held Shares, CAPEX, and Leverage are not significant, although they have their predicted sign. We also employ firm fixed effects, and our main results remain the same (Colum (4)) (We performed Hausman test and the null hypothesis is rejected, ($\chi 2$ (1) = 35.29, $p = 0.000$), which means that the fixed-effects estimator should be employed.).

### 4.2.2. The Impact of Board Gender Diversity on Firm Performance: Robustness Checks

In the following section, we perform additional analyses in order to check the sensitivity of our findings.

### 4.2.3. Alternative Proxies for Firm Performance and Board Gender Diversity

We assess the robustness of our previous results to using alternative proxies for firm performance and board gender diversity. We first replaced ROA with three alternative proxies: *(i)* Tobin's Q as a market-based measure of performance; *(ii)* return on sales (ROS) as argued by Liu et al. [59] that "compared to ROS, ROA could be downward biased due to the occasional seasoned equity offerings (SEOs) and the subsequent assets escalations"; and *(iii)* return on equity (ROE). Models (5)–(7) of Table 5 report the results and as we can see, the coefficient of BGD still significantly positive at the less than 1% level. Secondly, we used the percentage of female executive members (ExMemGD (%)) and the percentage of female managers to the total number of managers (WomMangScore) as different measure for BGD, and as shown in Model (8)–(9) of Table 4, BGD remain significant and positive (In untabulated tests, we checked the impact of cultural diversity, measured by "The percentage of board members that have a cultural background different from the location of the corporation headquarters and the percentage of senior executives that have a cultural background different from the location of the corporation headquarters", on the firm value, and we found a positive and significant relationship between them.). This result again confirms our H1.

### 4.2.4. Addressing Endogeneity Concerns

There are three main sources of endogeneity: *(i)* omitted variables that are correlated with some independent variables and error terms; *(ii)* reverse causality; and *(iii)* variable measurement error. In our case, we use different approaches to deal with endogeneity concerns. First, we include additional firm- and country-level controls to check that our previous results are not biased due to omitted correlated variables. With reference to the existing literature, we re-run our regression model and we introduced individually (1) Loss variable as a dummy variable that takes the value of 1 if net income before extraordinary items is negative and 0 otherwise to capture the company's losses; (2) the Beta firm as a risk measure; and (3) the inflation rate (Inflation) to capture the macroeconomic conditions. As shown in columns (1)–(3) of Table 6, we find that our main findings are still valid. Second, the decision of appointing woman to the board could be determined by the firm's characteristics, specifically its performance. In other words, the most successful and efficient firms could attract and engage the most qualified and competent women on their boards. Thus, to control for the simultaneity and potential reverse causality issue, we use the generalized method of moments (GMM) model and a two-stage least squares (2SLS) regression analysis. In the first-stage estimation of 2SLS regression (column (4) of Table 6), BGD is regressed on particular instruments. As the previous literature (e.g., Abou-El-Sood [33], we selected SECRules; board size (BoardSize); and the lag of firm performance and size (LagROA and LagSize, respectively) as instruments since they have been shown to determines BGD (It should be noted that for simplicity and for the specific purposes of this test we have used the sample of American firms only, since they dominate our sample firms.). For the second-stage regression, we use the predicted value of the BGD measure as our main independent variable to estimate firm performance. Columns (5)–(6) of Table 6 report the results of the GMM and 2SLS regressions. As shown, BGD continues to impact positively and significantly the firm value, which mean that our main findings are not driven by the endogeneity of firm performance. Third, we use propensity score matching (PSM) as alternative technique to verify the reverse causality issue. Indeed, we use this technique to determine a sample that includes all firms that have a policy regarding the gender diversity of its board (Policy Board Diversity) and a benchmark or control set of firms without policy regarding the gender diversity of its board. We first identify the reference sample (Treatment group) of all Canadian and US firms existing in the Asset4

ESG database that have a gender diversity policy (Asset4 ESG offers a filter named Policy Board Diversity that indicates if the company has a policy regarding the gender diversity of its board.). Then, we construct the matched sample of firms that do not have a gender diversity policy but have similar characteristics as the reference sample (Control group). This technique allows us to control for observable differences in characteristics between the reference and matched set of firms. Following the previous literature (e.g., Bortolotti et al. and Fernandes [68,69]), we identify matched firms based on the firm size, year and industry. We use the nearest-neighbor technique, which consists of choosing the firm without a gender diversity policy that is closest in terms of probability of being a firm with a gender diversity policy. The outcome of the PSM procedure is a sample of 9517 firm-year observations, with 7204 Treatment group observations and 2313 control group observations. Panel B of Table 4 presents the results of the PMS technique. The table compares the ROA of Treatment group firms with control group firms. As we can see, firms with a gender diversity policy experienced significantly (at less than 1% level) higher ROA relative to firms without a gender diversity policy by almost 13.4%. More importantly, column (7) in Table 6 reports results of regression using propensity score matching. The outcomes show that the presence of a gender diversity policy causes the firm performance to be increased by an average of 10.52%. This result is highly significant at the less than 1% level, which approves our main evidence. Fourth and finally, as employed by Imbens and Wooldridge [70], we refer to the differences-in-differences (DID) approach. This technique allows us to further estimate the effect of board gender diversity on firm performance by comparing the change in performance for the treatment group to the change in performance for the control group. To perform the DID regression, we need to have two groups: a Treatment and Control group as identified above, and we need to have a reference date to be able to compare the variation before and after this date for the two groups.

To do so, and in line with Abou-El-Sood [33], we referred to the date of Securities and Exchange Commission rules (SEC Rules) about the requirement of board gender diversity policy disclosure (For simplicity, we only search for firms operating in the USA since our sample is dominated by US firms.). The SEC Rules were introduced in December 2009 but became effective in 2010 (https://www.sec.gov/rules/final/2009/33-9089.pdf (accessed on 10 July 2022)). Therefore, DID estimation allows us to compare the change in firm performance between our Treatment and Control group before and after the implementation of the SEC rules. Model (8) of Table 6 reports the results of DID estimation. Our variable of interest is Policy Board Diversity $\times$ SECRules. Policy Board Diversity refers to a dummy variable that takes the value 1 if the company has a policy regarding the gender diversity of its board (Treatment group) and 0 if not (Control group), whereas SECRules is a dummy variable equal to 1 if the specific year is part of the period when the SEC rule on board-gender policy disclosure is becoming effective (2010–2020) and 0 otherwise. As shown in model (8) of Table 5, our interaction variable of interest Policy Board Diversity $\times$ SECRules is positive and highly significant. The results confirm that the performance of firms with a board diversity policy after the adoption of the SEC rules is significantly higher to the performance of firms that: *(i)* have a board diversity policy but before the implementation of the SEC rules; *(ii)* do not have a board diversity policy and after the implementation of SEC rules; and *(iii)* do not have a diversity policy but before the implementation of SEC rules. The results of DID estimation confirm again the positive effect of board gender diversity on firm performance.

**Table 6.** Further robustness checks, alternative sample compositions and regression frameworks. This table reports regression results for the impact of gender diversity on the firm performance. The main sample includes 14,382 firm-year observations representing 4008 North American listed firms over the period of 2002–2020. Models (1), (2) and (3) report results when we include potential omitted variables. Models (4) and (5) are used for the instrumental variable regressions. Model (6) reports results for the generalized method of moments' regression. Model (7) reports results for regression using a propensity score-matched sample. Model (8) reports results using difference-in-difference estimation. Model (9) reports results using a sample of just Canadian firms. Model (10) reports results excluding the years 2016–2018 from the main sample. Appendix A reports all variables description. The superscript ***, **, and * denote statistical significance at the 1%, 5%, and 10% levels, respectively.

| | (1) | (2) | (3) | Instrumental Variables (2sls) | | (6) | (7) | (8) | (9) | (10) |
|---|---|---|---|---|---|---|---|---|---|---|
| | Loss | Beta | Inflation | First Stage DepVar: BGD | Second Stage DepVar: ROA | GMM | PMS Regression | DID Estimation | Excluding USA | Excluding 2016–2018 Period |
| BGD | 0.021 ** (2.497) | 0.07 ** (2.572) | 0.024 ** (2.087) | | 0.470 *** (0.000) | 0.306 *** (0.001) | | | 0.095 *** (7.572) | 0.067 *** (3.319) |
| Additional Controls | −17.83 *** (−25.568) | −2.363 *** (−5.579) | −1.039 *** (−3.386) | | | | | | | |
| SECRules | | | | 1.984 *** (0.000) | | | | | | |
| BoardSize | | | | 0.943 *** (0.000) | | | | | | |
| LagROA | | | | 0.0459 *** (0.002) | | | | | | |
| LagSize | | | | 0.000 (1.000) | | | | | | |
| Policy Board Diversity | | | | | | 10.52 *** (0.000) | | 3.01 *** (0.000) | | |
| SECRules | | | | | | | | 0.849 *** (0.000) | | |
| Policy Board Diversity × SECRules | | | | | | | | 2.154 *** (0.000) | | |
| Gov | 0.022 *** (4.216) | 0.078 *** (17.015) | 0.053 *** (7.17) | | 0.00744 (0.538) | 0.00416 (0.727) | 0.0409 *** (0.001) | | 0.015 *** (3.934) | 0.017 ** (2.439) |
| CEO duality | −2.65 *** (−5.52) | −3.285 *** (−5.599) | −3.215 *** (−9.761) | | −3.764 *** (0.000) | −2.223 *** (0.006) | −8.555 *** (0.000) | | −2.613 *** (−9.392) | −1.771 *** (−3.4) |
| Held Shares | 0.021 *** (4.379) | 0.026 *** (2.87) | 0.009 (1.087) | | 0.0300 *** (0.003) | 0.0191 * (0.060) | 0.0658 *** (0.000) | | −0.001 *** (−8.753) | −0.015 * (−1.808) |
| CAPEX | 2.517 (1.037) | 20.846 *** (8.83) | 4.72 (1.43) | | 11.58*** (0.000) | 0.812 (0.788) | 0.000 (1.000) | | −11.21 *** (−2.199) | −4.776 ** (−2.289) |
| Leverage | 0.07 ** (2.377) | 0.075 (1.007) | −0.012 (−0.353) | | −0.0266 (0.534) | 0.133 *** (0.003) | 0.000 (1.000) | | 0.031*** (1.614) | 0.025 (0.646) |
| Age | 0.01 *** (3.719) | 0.069 ** (1.989) | 0.046 *** (12.931) | | 0.0320 *** (0.000) | 0.0224 *** (0.000) | 0.101 *** (0.000) | | 0.03 *** (5.413) | 0.019 *** (8.977) |
| Z-Score | 0.515 *** (19.747) | 0.213 *** (7.491) | 0.556 *** (21.134) | | 0.537 *** (0.000) | 0.754 *** (0.000) | 0.955 *** (0.000) | | 0.443 *** (3.904) | 0.58 *** (8.681) |
| Size | 2.059 *** (24.119) | 4.544 *** (24.033) | 2.754 *** (35.391) | | 2.334 *** (0.000) | 0.820 *** (0.000) | 7.538 *** (0.000) | | 1.03 *** (3.370) | 0.841 *** (4.726) |
| GDP | −0.93 (−1.499) | −2.773 *** (−3.464) | −0.953 (−1.424) | | 22.96 *** (0.000) | 8.786 * (0.082) | −1.487 (0.567) | | 0 (0) | 0.767 *** (10.747) |
| _cons | −42.064 *** (−24.198) | −96.889 *** (−32.739) | −60.426 *** (−32.939) | −0.869 (0.614) | −60.52 *** (0.000) | −21.43 *** (0.000) | −159.8 *** (0.000) | 0.487 ** (0.048) | −11.974 *** (−2.265) | −11.424 *** (−3.28) |
| Observations | 14,382 | 4373 | 14,382 | 21,703 | 13,081 | 13,081 | 2146 | 917 | 1276 | 6609 |
| Adj R * | 0.324 | 0.254 | 0.201 | 0.152 | 0.212 | 0.112 | 0.279 | 0.240 | 0.132 | 0.132 |
| Sig. | (0.000) | (0.000) | (0.000) | (0.000) | (0.000) | (0.000) | (0.000) | (0.000) | (0.000) | (0.000) |
| Year Fixed Effects | YES | YES | YES | YES | YES | YES | YES | NO | YES | YES |
| Industry Fixed Effects | YES | YES | YES | YES | YES | YES | YES | NO | YES | YES |
| Country Fixed Effects | YES | YES | YES | YES | YES | YES | YES | NO | YES | YES |

## 4.2.5. Other Robustness Checks: Alternative Sample Compositions

In this section, we test the sensitivity of our findings on various sample compositions. As mentioned before and reported in the descriptive statistics table (Panel A of Table 1), US firms dominate our sample (91.13%). To check whether our results are affected by US firms,

we perform our regression excluding US firms. Model (9) of Table 5 shows the results. As we can see, our previous inferences continue to hold even when we kept only Canadian firms. Moreover, as reported in the descriptive statistics, more than 54% of our sample observations belong to the period from 2016 to 2019 (Panel C of Table 1). Therefore, to ensure that our findings are not driven by the economic characteristics belonging to this period, we rerun our model excluding the period from 2016 to 2019. As shown in the model (10) of Table 5, the results remain the same, supporting the positive impact of board gender diversity on firm value.

### 4.3. Board Gender Diversity and Performance: The Mediating Role of Intellectual Capital Efficiency

4.3.1. The Impact of Intellectual Capital Efficiency on Firm Performance

Model (2) of Table 7 shows the results of OLS regression of intellectual capital efficiency (VAIC) on return on assets (ROA). As we can see, VAIC positively affects the ROA, confirming the results of Clarke et al., Chen et al., Bollen et al., and Cohen and Kaimenakis [10,43,70,71], among others. These findings are aligned with our H3, and they are maintained when we use innovation capital efficiency (SEC), physical capital efficiency (CEE) or human capital efficiency (HCE) as measures of intellectual capital efficiency (see columns (2)–(5) of Table 7). In addition, when using other proxies for firm performance, such as ROE or ROS, the findings still hold.

**Table 7.** Gender diversity, intellectual capital efficiency and firm performance. This table presents results of various OLS regressions. Model (1) reports results of the impact of BGD on VAIC. Model (2) reports results of the impact of VAIC on ROA. Mode (3) reports results of the impact of SEC on ROA. Model (4) reports results of CEE on ROA. Model (5) reports results of HCE on ROA. Appendix A reports all variables description. The superscript ***, **, and * denote statistical significance at the 1%, 5%, and 10% levels, respectively.

| | (1): Dep Var: VAIC | (2): Dep Var: ROA | (3): Dep Var: ROA | (4): Dep Var: ROA | (5): Dep Var: ROA |
|---|---|---|---|---|---|
| | Indp Var: BGD | Indp Var: VAIC | Indp Var: SEC | Indp Var: CEE | Indp Var: HCE |
| BGD | 0.005 ** | | | | |
| | (2.529) | | | | |
| VAIC | | 0.696 *** | | | |
| | | (11.883) | | | |
| SEC | | | 1.233 *** | | |
| | | | (5.999) | | |
| CEE | | | | 15.291 *** | |
| | | | | (9.649) | |
| HCE | | | | | 0.566 *** |
| | | | | | (8.25) |
| Gov | 0.012 *** | 0.01 | −0.029 *** | −0.004 | 0.01 |
| | (6.757) | (0.955) | (−2.769) | (−.273) | (1.04) |
| CEO duality | −0.013 | −2.841 *** | −2.369 *** | −2.733 *** | −1.788 * |
| | (−0.014) | (−2.698) | (−2.712) | (−2.98) | (−1.876) |
| Held Shares | −0.003 | −0.001 | −0.025 | 0.003 | 0.009 |
| | (−0.655) | (−0.05) | (−0.771) | (0.154) | (0.649) |
| CAPEX | 4.123 *** | −5.375 | 0.973 | −2.919 | −2.709 |
| | (3.386) | (−0.935) | (0.271) | (−1.445) | (−0.52) |
| Leverage | 0.004 | −0.037 | −0.2 *** | −0.335 *** | −0.052 |
| | (0.157) | (−0.499) | (−2.802) | (−3.15) | (−0.857) |
| Age | 0.019 | 0.034 *** | 0.047 *** | 0.023 | 0.026 |
| | (1.356) | (3.044) | (6.036) | (1.438) | (1.263) |
| Z-Score | 0.084 *** | 0.514 *** | 0.36 *** | 0.416 *** | 0.522 *** |
| | (5.469) | (3.792) | (5.237) | (4.494) | (5.353) |
| Size | 0.059 | 2.454 *** | 3.456 *** | 2.603 *** | 1.533 *** |
| | (0.705) | (10.05) | (16.204) | (13.675) | (10.364) |
| GDP | −0.474 *** | −0.847 | −0.036 | −1.037 ** | −2.898 *** |
| | (−28.928) | (−1.391) | (−0.019) | (−2.314) | (−3.12) |
| _cons | 1.763 | −52.743 *** | −78.229 *** | −55.654 *** | −30.295 *** |
| | (0.76) | (−6.843) | (−8.954) | (−9.81) | (−7.601) |
| Observations | 3600 | 3425 | 1333 | 3392 | 2494 |

**Table 7.** *Cont.*

|  | (1): Dep Var: VAIC | (2): Dep Var: ROA | (3): Dep Var: ROA | (4): Dep Var: ROA | (5): Dep Var: ROA |
|---|---|---|---|---|---|
|  | Indp Var: BGD | Indp Var: VAIC | Indp Var: SEC | Indp Var: CEE | Indp Var: HCE |
| Adj R * | 0.033 | 0.245 | 0.341 | 0.353 | 0.191 |
| Sig. | (0.000) | (0.000) | (0.000) | (0.000) | (0.000) |
| Year Fixed Effects | YES | YES | YES | YES | YES |
| Industry Fixed Effects | YES | YES | YES | YES | YES |
| Country Fixed Effects | YES | YES | YES | YES | YES |

### 4.3.2. The Impact of Gender Diversity on Intellectual Capital Efficiency

Existing research documents that intellectual capital efficiency is determined by firm-specific characteristics and macroeconomic factors. We performed our model where the dependent variable is the intellectual capital efficiency coefficient (*VAIC*) and the independent variable of interest is *BGD*. As reported in Model (2) of Table 7 and in line with Nadeem et al. [61], board gender diversity positively and significantly affects the firm's intellectual capital efficiency. This result confirms our H2. Moreover, the results (untabulated for brevity) of using other proxies for board gender diversity (e.g., WomMangScore) as an independent variable approve our finding that board gender diversity increases the firm's intellectual capital efficiency.

### 4.3.3. The Mediating Role of Intellectual Capital Efficiency

We are interested in the effect of BGD on firm performance, but we have the suspicion that a portion of the effect might be mediated through intellectual capital efficiency (ICE). Therefore, to test the ICE mediating role, we use the structural equation model (SEM) with the maximum-likelihood estimator adjusted at the firm level. SEM includes all the hypothesized paths, in addition to any direct and indirect effects, to investigate ICE mediation. Table 8 presents the results of the mediating model, indicating the direct, indirect and total effect of ICE as a mediating variable in the relationship between board gender diversity (BGD) and firms' financial performance (ROA). Model (1) presents the results of the mediating role as measured by innovation capital efficiency (SEC), while Model (2) indicates the mediating effect of physical capital efficiency (CEE). Model (3) indicates the mediating role of human capital efficiency (HCE), and finally Model (4) reports results of the mediating role of total ICE as measured by value-added intellectual (VAIC), which represents the coefficient calculated by summing innovation capital efficiency, physical capital efficiency and human capital efficiency (i.e., VAIC = SEC + CEE + HCE).

As shown in the Model (1), Model (2), Model (3) and Model (4) of Table 8, BGD positively and significantly affects firm performance as measured by ROA. For example, as reported in Model (4) of Table 8, the direct effect of BGD on ROA is estimated to be 0.145. The effect is positive and highly statistically significant. Moreover, and in line with Clarke et al. [71], among others, the results (untabulated for brevity) indicate that the direct effect of VAIC on firm performance is positive and highly significant and is estimated to be 0.566. Likewise, the direct effect of BGD on VAIC is positive and statistically significant and is estimated to be 0.019.

More importantly, the total effect of BGD on ROA is estimated to be 0.155, which is greater than the direct effect (0.145). This means that there is part of the effect of BGD going through the mediator variable VAIC. Effectively, the indirect effect of BGD that passes through VAIC is estimated to be 0.01. This effect is relatively small, albeit statistically significant, confirming the mediating role of intellectual capacity efficiency between BGD and firm performance and confirming our hypothesis H4. The findings are robust when adding control variables in the mediating model (Model (5) of Table 8).

We further tested the robustness of our results reported in Table 9. We tested the results using other measures of performance, namely ROE, TobinQ and ROS. As shown in Model (1), Model (2) and Model (3), our results remain the same. The indirect effect of BGD is positive and statistically significant and estimated to be 0.018, 0.002 and 0.001

(see Model (1), Model (2) and Model (3) of Table 9, respectively), confirming the mediating role of ICE on the relationship between BGD and firm performance. Further, we use two other proxies for board gender diversity to test the sensitivity of our results. The first is the percentage of female executive members (ExMemGD (%)). The second proxy is the percentage of female managers to the total number of managers (WomMangScore). The results detailed in model (4) and Model (5) of Table 9 confirm our main finding. Mostly, the regression results (untabulated for brevity) after adding control variables to mediating models remain the same. Overall, based on the various sensitivity tests we have strong evidence that intellectual capital efficiency mediates the relationship between board gender diversity and firm performance.

**Table 8.** Results of Mediating Model. This table presents the results of the mediating role of intellectual capital efficiency in the relationship between BGD and firm performance using structural equation models. The table reports results on the total, direct and indirect effects. Appendix A reports all variables description.

| | Model 1: SEC | | | Model 2: CEE | | | Model 3: HCE | | | Model 4: VAIC | | | Model 5:VAIC *with Controls* | | |
|---|---|---|---|---|---|---|---|---|---|---|---|---|---|---|---|
| | *BGD and ROA* | | | *BGD and ROA* | | | *BGD and ROA* | | | *BGD and ROA* | | | *BGD and ROA* | | |
| | IE | DE | TE | IE | DE | TE | IE | DE | TE | IE | DE | TE | IE | DE | TE |
| BGD | 0.052 | 0.19 | 0.242 | 0.077 | 0.076 | 0.154 | 0.007 | 0.020 | 0.028 | 0.01 | 0.145 | 0.155 | 0.017 | 0.13 | 0.148 |
| | (0.000) | (0.000) | (0.000) | (0.000) | (0.001) | (0.001) | (0.009) | (0.059) | (0.012) | (0.036) | (0.000) | (0.000) | (0.005) | (0.000) | (0.000) |
| Gov | | | | | | | | | | | | | | | −0.003 |
| | | | | | | | | | | | | | | | (0.751) |
| CEO duality | | | | | | | | | | | | | | | −1.89 |
| | | | | | | | | | | | | | | | (0.113) |
| Held Shares | | | | | | | | | | | | | | | −0.009 |
| | | | | | | | | | | | | | | | (0.436) |
| CAPEX | | | | | | | | | | | | | | | −4.25 |
| | | | | | | | | | | | | | | | 0.469 |
| Leverage | | | | | | | | | | | | | | | −0.112 |
| | | | | | | | | | | | | | | | (0.003) |
| Age | | | | | | | | | | | | | | | 0.016 |
| | | | | | | | | | | | | | | | (0.018) |
| Z-Score | | | | | | | | | | | | | | | 0.469 |
| | | | | | | | | | | | | | | | (0.000) |
| Size | | | | | | | | | | | | | | | 2.448 |
| | | | | | | | | | | | | | | | (0.000) |
| GDP | | | | | | | | | | | | | | | −0.000 |
| | | | | | | | | | | | | | | | (0.908) |
| Observations | 2382 | | | 7158 | | | 6002 | | | 7221 | | | 3227 | | |
| Sig. | (0.000) | | | (0.000) | | | (0.000) | | | (0.000) | | | (0.000) | | |

Note: IE = indirect effect; DE = direct effect; TE = total effect, and *p*-values are shown between parentheses.

**Table 9.** Results of Mediating Model: *Robustness checks.* This table details results of robustness tests of the mediating role of intellectual capital efficiency in the relationship between *BGD* and firm performance using structural equation models. The table reports results on the total, direct and indirect effects. Appendix A reports all variables description.

| | Model 1: VAIC | | | Model 2: VAIC | | | Model 3: VAIC | | | Model 4: VAIC | | | Model 5: VAIC | | |
|---|---|---|---|---|---|---|---|---|---|---|---|---|---|---|---|
| | *BGD and ROE* | | | *BGD and Tobin-Q* | | | *BGD and ROS* | | | *ExMemGD (%) and ROA* | | | *WomMangScore and ROA* | | |
| | IE | DE | TE | IE | DE | TE | IE | DE | TE | IE | DE | TE | IE | DE | TE |
| BGD | 0.018 | 0.516 | 0.535 | 0.002 | 0.027 | 0.03 | 0.001 | 0.001 | 0.002 | | | | | | |
| | (0.056) | (0.000) | (0.000) | (0.000) | (0.000) | (0.000) | (0.043) | (0.000) | (0.003) | | | | | | |
| ExMemGD (%) | | | | | | | | | | 0.059 | 0.136 | 0.195 | | | |
| | | | | | | | | | | (0.018) | (0.000) | 0.002 | | | |
| WomMangScore | | | | | | | | | | | | | 0.03 | 0.003 | 0.033 |
| | | | | | | | | | | | | | (0.031) | (0.000) | (0.017) |
| Observations | 6990 | | | 886 | | | 863 | | | 3071 | | | 472 | | |
| Sig. | (0.000) | | | (0.000) | | | (0.000) | | | (0.000) | | | (0.000) | | |

Note: IE = indirect effect; DE = direct effect; TE = total effect and *p*-value between parentheses.

## 5. Discussion and Conclusions

During the past two decades, we have witnessed an increasingly intense pressure, coming from stakeholders, on organizations to have more representation of women on the corporate board (CB) and a growing interest from academic research on the effects of board gender diversity on the financial performance and corporate governance. However, evidence that board diversity benefits firms is mixed and inconclusive. Moreover, the mechanisms by which BGD influences firm performance are always ambiguous.

This study extends previous research on board gender diversity and financial performance. It is also a part of our efforts to understand the mechanisms that explain the relationship between gender diversity on board and organizational performance. Furthermore, in the era of the knowledge economy, we wanted to understand the role of intellectual capital in the dynamic relationship between gender diversity and financial performance. More specifically, and in line with our paper published in 2020 [31] on the role of social responsibility, this article introduces a new variable: intellectual capital. Because the inconsistency in results is often explained by a variety of scales measuring intellectual capital or by the heterogeneity of the sample, the present study used the widely accepted VAIC measure, which allows us to compare our results with those of other studies. Using univariate analysis and OLS regression to study the impact of BGD on firm performance and SEM to test the mediating role of VAIC, the research results are promising in several respects. On the one hand, the study confirms the importance of gender diversity in supporting the financial performance of organizations (hypothesis H1 supported), and on the other, the mediating role of intellectual capital in this relationship (hypotheses H2, H3 and H4). Thus, it seems that part of the effect of diversity on performance is mediated by intellectual capital. This result is consistent with Hambrick and Mason's [39] upper-echelon theory, which states that the results of any organization depend (in part) on its internal characteristics, and in our case, the composition of the board of directors with respect to gender diversity.

The results of our study support the resource-based theory perspective since they reveal that intangible assets contribute to the profitability of organizations in two ways. The culture of diversity materialized by the representation of women on the corporate board and the strategic choices with regard to intellectual capital materialized by the efficiency of intellectual capital and its effects on financial performance. Moreover, because there is a direct and indirect effect (through intellectual capital) of gender diversity, it seems that the effect of diversity is not only limited to choices making intellectual capital more efficient (effect indirect), but to other effects that the study identifies with the direct impact on performance. In other words, gender diversity contributes to financial performance and its effect exceeds intellectual capital. Our finding is also in line with upper echelon theory, which conceives organization as a reflection of its top managers, and in our study, the importance of intellectual capital both on the corporate board (gender diversity) and in the strategic orientations (intellectual capital). These results could be beneficial to both enterprises as well as the policy-makers that are more concerned in the role of corporate governance, particularly the board composition. Therefore, engaging female in BODs, allow the board to make strategic decisions that value intellectual capital (human capital and structural capital) and transform it into organizational performance.

Despite the importance of the results and the scope of their implications, especially the non-financial benefits of gender diversity such as the efficiency of intangible capital, some limitations should be noted. First, the size of the sample limits the possibility of testing a more complex model that would allow us to isolate the effect of intellectual capital as a whole, but also the isolated effect of each of its dimensions, human capital and structural capital. Moreover, because the sample is limited to North American firms, we cannot generalize the results to other countries or economies such as Europe or Asia. Finally, because the indirect effect through intellectual capital is not very high, we suspect the existence of other variables in the causal chain linking gender diversity and intellectual capital (e.g., independence, expertise and experience, etc.).

Finally, we encourage researchers to explore other avenues of research to further document and understand the role of gender diversity in BODs. For example, it would be interesting to test the circularity of the relationship between intellectual capital and performance, as well as the role of gender diversity in the orientation of investments towards human capital and structural/innovation capital.

**Author Contributions:** Conceptualization, Z.O. and J.B.M.; Methodology, Z.O. and J.B.M.; Software, Z.O. and J.B.M.; Formal analysis, Z.O. and J.B.M.; Investigation, Z.O. and J.B.M.; Data curation, Z.O., J.B.M. and S.A.; Writing—original draft, Z.O. and J.B.M.; Writing—review & editing, S.A.; Project administration, S.A. All authors have read and agreed to the published version of the manuscript.

**Funding:** This research received no external funding.

**Conflicts of Interest:** The authors declare no conflict of interest.

## Appendix A

**Table A1.** Definition of variables.

| Variable | Expected Sign | Description | Source |
|---|---|---|---|
| ROA | | Return on assets (profitability ratio) calculated as "the ratio of earnings before interest, taxes, depreciation and amortization (EBITDA) to total assets." | Authors' calculation based on data from Worldscope |
| BGD | ? | Percentage of women on the board. | Asset4 ESG |
| CEE | + | Physical capital efficiency measured as the "Value added/capital employed" (where capital employed is the total firms' capital). | Authors' calculation based on data from Worldscope |
| HCE | + | Human capital efficiency measured as "Value added/total personnel cost". | As above |
| SCE | + | Innovation capital efficiency measured as "Value added/structural capital" (structural capital is R&D). | As above |
| VAIC | + | Value-added intellectual. Coefficient Human capital efficiency plus structural capital efficiency plus financial capital efficiency. | As above |
| Size | + | Firm size calculated as the natural log of the total assets in millions of US dollars. | As above |
| Leverage | − | The firm leverage, which is calculated as "the ratio of total debts to total assets". | As above |
| Gov | + | The governance quality rating of the company. | Asset4 ESG |
| CEO duality | − | Dummy variable that takes the value 1 if the CEO is the Chairman person; 0 otherwise. | Asset4 ESG |
| Held Shares | ? | Represents shares held by insiders. In Worldscope, "Closely held shares comprise (1) shares held by insiders, including senior corporate officers, directors, and their immediate families, (2) shares held in trusts, (3) shares held by another corporation (except shares held in a fiduciary capacity by financial institutions), (4) shares held by pension/benefit plans, and (5) shares held by individuals who hold 5% or more of shares outstanding". | Worldscope |
| CAPEX | + | The ratio of capital expenditures to total assets. | Authors' calculation based on data from Worldscope |
| Age | + | The firm age. | As above |
| Z-Score | + | "Altman's (1968) Z-score = $6.56 \times$ (working capital/total assets) + $3.26 \times$ (retained earnings/total assets) + $6.72 \times$ (earnings before interest and taxes/total assets) + $1.05 \times$ (book value of firm/book value of total liabilities)". | As above |
| GDP | + | The GDP growth. | IMD World Competitiveness Center Database |
| Policy Board Diversity | + | Dummy variable that takes the value 1 if the company has a policy regarding the gender diversity of its board; 0 otherwise. | Asset4 ESG |
| ExMemGD (%) | + | "Percentage of female executive members". | Asset4 ESG |
| BCulD (%) | + | "The percentage of board members that have a cultural background different from the location of the corporation headquarters". | Asset4 ESG |
| ExCulD | + | "Percentage of senior executives that have a cultural background different from the location of the corporation headquarters". | Asset4 ESG |
| WomMangScore | + | "The percentage of women managers to the total number of managers." | Asset4 ESG |
| SECRules | ? | Dummy variable that takes the value 1 if the specific year is part of the period when the SEC rule on board-gender policy disclosure is effective (2010–2020); 0 otherwise. | Worldscope |
| BoardSize | + | "The total number of board members at the end of the fiscal year". | Asset4 ESG |

**Table A1.** *Cont.*

| Variable | Expected Sign | Description | Source |
|---|---|---|---|
| Loss | – | Dummy variable that takes the value 1 if net income before extraordinary items is negative in the current and prior fiscal year; 0 otherwise. | Authors' calculation based on data from Worldscope |
| Beta | + | Risk measure. | Worldscope |
| Inflation | – | The annual rate of inflation for the prior fiscal year. | IMD World Competitiveness Center Database |
| ROS | | Return on sales (operating profit) measured as "the ratio of earnings before interest and taxes (EBIT) to net sales". | Authors' calculation based on data from Worldscope |
| ROE | | Return on equity ratio calculated as the ratio of net income by its shareholder's equity. | As above |
| TobinQ | | "TobinQ calculated as the book value of total assets plus the market value of equity minus the book value of equity divided by total assets". | As above |

## Appendix B

**Table A2.** Sample selection.

| Steps | Tasks | Outcomes |
|---|---|---|
| Step 1: | Obtain board gender diversity information from *Asset4 ESG* database for all American and Canadian public firms. As a first filter, we only keep firms that have adopted board gender policy. | 27,053 firm-year observations covering the period from 2002 to 2021 (23,522 observations about 3556 American firms and 3532 observations about 454 Canadian firms) |
| Step 2: | Collect performance and other financial data: Match board gender diversity data with financial data from *Worldscope* database. | 15,986 firm-year observations covering the period from 2002 to 2020 (14,362 observations about 3556 American firms and 1823 observations about 452 Canadian firms). |
| Step 3: | Collect macroeconomic data: Match board gender diversity and financial data with economic indicators from *IMD World Competitiveness Center* database. | Main sample: 14,382 firm-year observations covering the period from 2002 to 2020 (13,106 observations about 3556 American firms and 1276 observations about 452 Canadian firms). |
| Step 4: | Collect and manual calculation of intellectual data: Match our main sample with intellectual data (VAIC). | 3425 firm-year observations covering the period from 2002 to 2020 (2570 observations about 3556 American firms and 855 observations about 452 Canadian firms). |

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
