# Peer review of "Corporate Governance and Financial Performance: The Interplay of Board Gender Diversity and Intellectual Capital"

_sustainability, doi:10.3390/su142215232_

Round 1

Reviewer 1 Report

  I have read your paper "Corporate Governance And Financial Performance: The Inter-play Of Board Gender Diversity And Intellectual Capital". I find that the presentation of the article and the idea are good however, I have the following minor concerns: 

TThe first two paragraphs in introduction have no citations

2.    Could you justify why the sample is from North American?

3.    You need to explain the sample selection, population, data filter, and the final sample. It is better to include a table or figure to show the sampling process.

4.    Your equation model has some types. You should also number your equations.

5.    Since you have hypotheses, you should discuss your findings with a ling to your hypotheses.

6.    The discussion and linking of the results with prior studies is very weak and needs to be more discussed. The results should be discussed in light of the findings from prior studies.

7.       I suggest to show Husman Test value to support your fixed or random effect models.

8.       The references need to be revised both intext and reference list.

Reviewer 2 Report

Abstract

ü  The abstract should include the problem statement, objective, methodology, findings, and recommendations. Please the author(s) should adjust the abstract.

Introduction

ü  The background is well written but lacks prevalent and specific issues about financial performance and corporate governance.

ü  The author(s) should provide theoretical and empirical arguments that show the relationship among the variables in the context.

ü  The author(s) should also discuss the link among Corporate Governance, Financial Performance, Gender Diversity, and Intellectual Capital. The integration of these variables should be justified in the background.

ü  There is no flow in the background.

ü  The objectives of the paper should be clearly defined and directed towards solving the problem statement.

ü  Many intext citations were not included in the reference list.

ü  The author(s) should include more recent citations. Most of the citations are old.

Literature Review

ü  The author(s) should clearly define the different concepts with empirical clarifications

ü  The author(s) should also add major theories relating to the work and link the theories to the issues raised in the paper

Methodology

ü  The methodology needs a lot of justifications. Kindly be specific with the population, sample, sampling techniques and research instrument.

ü  The author(s) should state the reason for using the return on assets (ROA) and structural equation model (SEM)?

Data Analysis

ü  All the figures and tables should be appropriately linked and labelled

ü  The paper lacks thorough discussions of findings

ü  Proper justification be giving to the use of SEM

Conclusion

ü  The conclusions are vague and not supported by the results.

ü  The author(s) need to ensure that all references are relevant to the contents of the
manuscript.

ü  Kindly include recommendations and policy implications

ü  This paper needs proper editing.

Round 2

Reviewer 2 Report

ü  The author should have a section for recommendations 
